# Cryopreservation by Directional Freezing and Vitrification Focusing on Large Tissues and Organs

**DOI:** 10.3390/cells11071072

**Published:** 2022-03-22

**Authors:** Amir Arav

**Affiliations:** A.A Cash Technology, 59 Shlomzion Hamalca, Tel Aviv 62266, Israel; arav.amir10@gmail.com

**Keywords:** directional freezing, vitrification, organ, tissue, ice crystals, cryopreservation

## Abstract

The cryopreservation of cells has been in routine use for decades. However, despite the extensive research in the field, cryopreservation of large tissues and organs is still experimental. The present review highlights the major studies of directional freezing and vitrification of large tissues and whole organs and describes the different parameters that impact the success rate of large tissue and organ cryopreservation. Key factors, such as mass and heat transfer, cryoprotectant toxicity, nucleation, crystal growth, and chilling injury, which all have a significant influence on whole-organ cryopreservation outcomes, are reviewed. In addition, an overview of the principles of directional freezing and vitrification is given and the manners in which cryopreservation impacts large tissues and organs are described in detail.

## 1. Introduction

While more than 70 years have passed since the first successful attempt at the cryopreservation of cells, whole-organ freezing is still in its initial phases. In the pioneering works of Audrey Smith [1], who first performed many of the early freezing experiments, hamster hearts perfused with 15% glycerol and exposed to −20 °C resumed a rhythmic heart beat after thawing. However, recovery was not achieved after freezing to lower temperatures [1]. The success of large-tissue and organ cryopreservation by slow freezing or vitrification has been limited by heat and mass transfer [2]. Heat transfer from the organ (i.e., cooling) and to the organ (i.e., heating) primarily occurs via convection (i.e., through gas or liquid material), which always begins from the outer parts of the organ and progresses toward its center. Additionally, heat transfer by convection is slower relative to heat transfer by conduction (i.e., through solid material). The variable cooling rates across large-tissue and whole-organ samples, dictated by their geometry (surface, depth, and volume) and heat conductivity, pose significant challenges to attempts at cryopreservation.

At present, directional freezing (DF) and vitrification are the two main methods used for the long-term cryopreservation of whole organs. Vitrification, a process of solidifying water without the formation of ice crystals, requires high viscosity, high cooling/warming rates and small sample volumes [3,4,5,6,7,8,9,10,11,12,13]. Directional freezing is a method in which the sample is frozen while moving through a temperature gradient at a constant velocity [5]. Both techniques bring sample temperatures to below −80 °C and enable control of the speed of crystallization—during DF by changing the velocity of ice crystals, and during vitrification by changing the cryoprotectant (CP) concentration (viscosity) and the cooling rate (∆T). DF has been successfully applied to cryopreserve a variety of cells and tissues [7,8,9,12,13]. We and others have shown that DF is both highly reproducible and associated with high survival rates compared to non-directional slow freezing methods and other reported cell cryopreservation methods [5,6,7,8,9]. It has been applied to freeze whole ovarian slices and whole ovaries of sheep [14,15,16], which, after short-term storage in liquid nitrogen (LN), were thawed and transplanted back to the same animal [6]. Furthermore, 6 years post transplantation, ovaries were still functioning [15]. DF was also successfully applied to preserve human ovaries [16]; pig livers, which were transplanted to the same animal and produced bile; [17] and a rat heart, which resumed its function (i.e., rhythmic heart beat) following cryopreservation at −8 °C [18]. Vitrification has been utilized to cryopreserve the kidneys, whole ovaries [19,20], whole limbs [21], blood vessels [22,23], cornea and valves [24,25], cartilage [26]. Recently, successful transplantation of a rat hind limb after cryopreservation by DF and vitrification was reported [21,27].

## 2. Principles of Directional Freezing (DF) and the Effect of Ice Crystal Propagation Velocity

DF is an alternative approach for preserving large organs, based on the physical concept that enables precise adjustment of temperature gradients to achieve an accurate and uniform cooling rate throughout the entire tissue [5]. This method has solved problems associated with heat release by increasing heat transfer through the entire organ [13,14,15,16,17,18]. DF was first introduced to the field of cryobiology with the design of a cryo-microscope [28] and it was then further modified, resulting in a new freezing device named ‘Multi-Thermal-Gradient’ (MTG) [5]. The device comprises a series of heat-conductive blocks (usually made of brass or aluminum) arranged in a line, with a straight track running through the blocks. Different temperatures (T1, T2, T3, and T4, see Figure 1) can be preset along the blocks, thereby imposing a temperature gradient along and between the blocks (G1, G2, and G3). The blocks are separated by a gap and the temperature of the block on one side of the gap (T1) is above freezing point, while on the other side of the gap (T2), it is below freezing point, thereby imposing a temperature gradient across the gap (G1). Biological samples to be frozen or thawed are placed inside tubes and pushed along the track at a certain velocity (V), which can also be altered. The freezing rate is dictated by specific protocols, which differ by sample size and intended use. To achieve the same cooling rate in the entire organ, the velocity should remain slower than the heat transfer velocity between the center of the tissue/organ and the conductive material of the blocks (Figure 1). The MTG can generate a linear thermal gradient within the organ/tissue to be cooled. Heat transfer can then be modified in a directional way, in accordance with the heat transfer in the sample (i.e., as slow as 0.04 to 0.06 mm/s). The speed of movement of the organ within the thermal gradient should not exceed this velocity in order for heat to remain directional. In this way, heat transfer will propagate perpendicular (Figure 1) to the direction of the movement of the organ, and the thermal history (cooling rate) will be identical for each point in the organ.

When heat transfer is perpendicular to the direction of movement, the velocity (V) is slower than the speed at which the heat is removed from the center of the sample toward its periphery. Thus, latent heat is removed toward the direction of the conductive block, which minimizes the effect of heat on the frozen material (exothermic heat release) [17]. In addition, it minimizes supercooling of the tissue; in DF, ice formation (nucleation) is determined by the solution’s freezing point temperature, and therefore supercooling is avoided completely [12]. In summary, DF, which involves the very slow movement of a whole organ through a linear temperature gradient, enables a precisely controlled cooling rate, ice crystal propagation velocity and a rapid release of latent heat. However, when freezing cells in suspension, it has been found that the velocity (V) should be increased to allow ice crystal growth in the opposite direction of the movement through the thermal gradient (see Figure 1). Indeed, cryo-microscopy observations showed that sperm cells surviving DF could be described with an inverted U-shape curve: at a very slow velocity (0.04 mm/s), ice grew slowly and in a planar form, which killed all cells; at a higher velocity (1.5 mm/s), ice crystals formed secondary branches and survival was increased; at a very high velocity (4 mm/s) DF did not occur and survival decreased [29].

## 3. The Evolution of Vitrification

Vitrification is a physical process by which liquid is transformed into a solid of amorphous glass. The transformation is due to the extreme elevation in viscosity from 1–10 poise to ~10^13^ poise, at which point the liquid is considered to have reverted to a glassy or vitreous state [30]. Vitrification avoids the process of crystal nucleation followed by crystal growth, thereby enabling a high-viscosity liquid to maintain its disordered molecular arrangement. Vitrification can be regarded as “the extreme case of supercooled water” [31,32] and was summarized by Luyet, who reported that “some of the oldest investigations on subcooling were made by Gay Lussac (1836) who observed that water can be subcooled to −12 °C when it is enclosed in small tubes” [33]. Already in 1858, after spraying droplets of water less than 0.5 mm in diameter on a dry surface, Johann Rudolf Albert Mousson observed that the smaller the drops were, the longer they stayed subcooled [34]. Yet, apart from the importance of volume in the achievement of supercooling, it was soon understood that the cooling velocity and the concentration of the supercooled or supersaturated [35] solutions, among other factors, might influence crystallization rates. As Luyet wrote: “To avoid freezing, the temperature should drop at a rate of some hundred degrees per second, within the objects themselves” and “The only method of vitrifying a substance is to take it in the liquid or gas state and cool it rapidly so as to skip over the zone of crystallization temperatures in less time than is necessary for the material to freeze. Therefore, one should considerate the crystallization velocity [36]. It is evident that when the crystals grow faster one must traverse the crystallization zone more rapidly if one wants to avoid crystallization” [33]. Notably, the currently used and most effective cooling rate for oocyte and embryo vitrification is in the range of 100 degrees per second, [37] as proposed by Luyet 80 years ago.

The first important, successful attempt at vitrification to preserve biological samples was made in 1968, when human erythrocytes were vitrified in rapidly cooled, ~5.3 molar glycerol, and remained intact after rewarming [38]. The first successful recovery of living nucleated cells (mouse embryos) from a liquid-nitrogen temperature was reported by Rall and Fahy in 1985 [39]. This stimulated research on the cryopreservation of complex and massive tissues and of organs [40,41,42,43,44,45,46]. Additional reviews of the history of biological vitrification are available elsewhere [12,37,44,47,48,49].

Pegg reported that a gradual introduction of a solution containing 55% *v*/*v* DMSO while reducing the temperature reduced the toxicity of this high-concentration cryoprotectant [50]. Indeed, he reported on the successful preservation of guinea pig uteri in a supercooled state at −79 °C (the temperature of subliming dry ice) with subsequent function after rewarming and removal of DMSO. Yet, this method poses an increased risk of chilling injury, as shown in cells exposed to low temperatures for prolonged periods of time [51,52]. Following this work, Kemp et al. reported on the cryopreservation of rat, cat, and dog kidneys by perfusing them with 55% DMSO at temperatures down to −40 °C, later storing them at −79 °C [53]. Rapatz reported on the successful vitrification of frog hearts with EG of 11 M (EG) [54,55].

### Modern Vitrification

Fahy was the first to propose the extension of deep supercooling down to glass transition temperatures (Tg) [55,56]. The Tg (glass transition temperature) of most vitrification solutions is at the change from a liquid to a solid state, which occurs near –120 °C (±10 °C) [57]. This method was motivated by the need to prevent thermal stresses which lead to fractures in large tissues and whole organs [21,58]. Fracturing is also responsible for devitrification [48], which involves ice nucleation and crystallization upon the release of heat via the fractures. These, in turn, alter tissues, organs, and even cellular systems [58,59,60,61,62,63,64,65]. The probability of fractures (Pfrac) can be expressed with the following equation [66]:Pfrac = η × CR(orWR) × V,
where η is the viscosity, CR and WR are the cooling or warming rates, respectively, and V is the volume. As expressed in this equation, the risk of fractures of small biological samples can be eliminated by reducing the volume of the drops to a minimum and by decreasing the viscosity (CP concentration) (MDS technique), as will be described next. Unfortunately, these solutions are not possible for large tissues and organs, as heat transfer is usually achieved by convection, which highly limits the CR and WR in large tissues and organs. Therefore, the only option is the use of a high concentration of CPs, which affects chemical toxicity and osmotic stress [67,68,69,70,71]. These can be reduced by the introduction of additives, such as sugars, macromolecules, polymers such as polyvinyl alcohol (PVA) of polyglycerol (PGL), antifreeze proteins, ice blockers or ice nucleating agents [72,73,74]. As described earlier, heat transfer is limited when using liquid nitrogen. We and others have demonstrated the benefit of liquid nitrogen slush (−210 °C) in increasing the cooling rate extensively [75,76,77]. Recent studies have shown the benefit of electromagnetic irradiation [78,79] and, more recently, of magnetic nanoparticle warming [80,81,82] (nanowarming) and warming with metal forms [83,84] to achieve faster and more uniform tissue heating during recovery from the vitrified state.

## 4. Factors Effecting Oocyte and Embryo Vitrification

As described, vitrification requires high viscosity, high cooling/warming rates and a small sample volume [85,86,87]. The probability of vitrification (Pvit) can be described with the following equation [12,13]:Pvit = CR(WR) η 1/V(1)

In the past, slow freezing was the method used to cryopreserve oocytes and embryos. Freezing requires nucleating factors that can induce ice crystal growth, either spontaneously as the temperature declines, or deliberately by inducing seeding [12]. Ice nucleation occurs spontaneously in a solution [21], and following shrinkage of the cells, the smaller the volume is, the lower is the chance that intracellular nucleation and crystallization occur [12]. When temperatures decline, the viscosity of the solution increases, its freezing point decreases and its glass transition temperature increases [2]. On the basis of this principle, it is possible to avoid freezing by increasing the viscosity (i.e., CP concentrations) and the cooling rates while decreasing sample volume [12,76,77]. Historically, vitrification efforts focused on increasing viscosity by adding high concentrations of CPs [39,40]. However, these concentrations are damaging due to chemical toxicity or osmotic stress [68,69]. Subsequently, attempts have been made to increase CRs and reduce CP concentrations, while rapid cooling rates were shown to allow a reduction in CP concentration and avoid chilling injury [75]. Many of the successful methods today involve plunging oocytes and embryos directly into LN in what is referred to as an open system [88]. However, this approach poses risks of contamination and cross-contamination [89,90,91]. These risks were solved by introduction of the CLAIR, a device which produces clean liquid air [92]. A third approach, proposed 30 years ago, reduces the sample volume to enable a reduction in CP concentration and an increase in the cooling rate [12,76]. This approach, termed Minimum Drop Size (MDS), addresses the three requirements for successful vitrification. Currently, this method is successfully applied worldwide for oocyte and embryo vitrification [93].

The nucleation rate of intracellular ice crystals is a function of temperature and cytoplasm composition [94].

The classic theory claims that a stable ice nucleus forms by random clustering of water molecules [2,12]. Therefore, nucleation is a statistical occurrence by nature, and thus, the more water molecules are present, the higher are the chances for nucleation to occur. Intracellular viscosity, which is dictated by water, salt and CP content, affects both nucleation rates and crystal growth rates [94]. As explained, the key factors in preventing nucleation are sample volume and intracellular viscosity. Adrian et al. showed that a thin (<1 µm) layer of pure water or dilute aqueous solutions resulted in vitrification upon immersion in liquid ethane or propane [95,96]. In the past, the success of vitrification was mainly attributed to the high cooling rate [35], however, this is an incomplete explanation, since once an ice nucleus has formed, the ice crystal propagation velocity is extremely high (unpublished study). As mentioned above, the chance for ice nucleation to occur correlates with sample volume [12,13]. Therefore, in a large volume, the ice propagation velocity (**u**), according to the Turnbull equation [97,98], is inversely correlated with the viscosity (η) and directly proportional to the function of the system’s supercooling (ʄ(ΔTr)), and can be plotted as follows:(2)$$u\;≌\;\frac{\mathrm{Ku}}{\eta }ʄ\left(\Delta \mathrm{Tr}\right)$$
where Ku is a constant determined by the designed model [97,98]. Ice crystal growth in water occurs at a velocity of **u** = 0.158∆T^1.69^ cm/s. Although the Turnbull equation was presented many years ago, to our knowledge, information on the speed of ice crystal propagation in vitrification solution containing different concentrations of CPs remains to be investigated.

## 5. Limitations and Challenges

Successful vitrification depends on several factors which affect cryoprotectant toxicity, nucleation, crystal growth, and chilling injury. In our experience (unpublished), we found that ice propagates in 10% (*v*/*v*) DMSO solutions at a velocity of 1 mm/s, if we calculate the CR that is needed to avoid crystal growth for ∆T of 60 °C (from −40 °C to −100 °C) for a vitrification solution containing 10% DMSO or EG, which is the oocyte’s intracellular CP concentration following exposure to an equilibration solution. Taken together, we found that the ice grew to 10 µm within 0.001 s. Growth by 10 µm was evaluated based on the finding that a proportion of 16% of intracellular ice crystals is detrimental to the oocyte [99]. Therefore, a cooling rate of 3 million °C/min is needed to avoid intracellular crystal growth to 10 µm at ∆T of 60 °C. Karlsson, who calculated that nucleation time for a large volume of solution containing 40% glycerol is in the range of nanoseconds, concluded that intracellular viscosity affects both nucleation and ice crystal growth rates [100]. Since the range of cooling/warming rates needed to avoid nucleation and crystallization in large volumes can be achieved, we can deduce that successful oocyte vitrification is achieved due to the combination of intracellular viscosity and a small volume, which affects the chance of nucleation, and is not due to the prevention of ice crystal growth.

In conclusion, for the successful vitrification of large tissues or organs, higher concentrations of CPs should be used. In parallel, the solution volume should be minimized to reduce potential toxicity and osmotic stresses. In addition, new technologies, such as warming with radiofrequency heating or nanowarming, can be implemented to increase cooling and warming rates. Another option is to combine DF and vitrification principles, by employing higher-CP-concentration formulations based on non-toxic CP, and increasing the CR while applying directional solidification principles.

## Figures and Tables

**Figure 1 cells-11-01072-f001:**
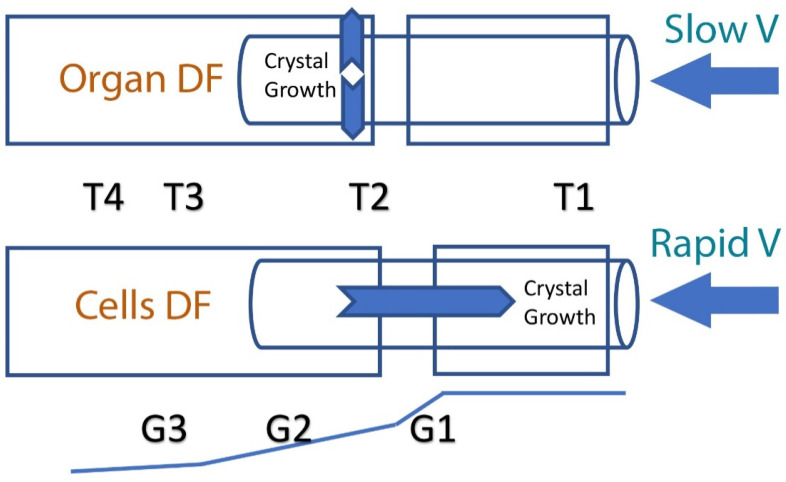
A schematic illustration of directional freezing of an organ and of cells. In the case of slow-velocity movement through the thermal gradients G1 and G2, heat transfer and ice crystal growth are perpendicular to movement. In cases of rapid velocity, heat transfer and ice crystal growth are in a direction opposite to that of the sample’s movement through the temperature gradient (G1). The cooling rate can be calculated according to this equation: B = G × V, where B is the cooling rate defined as ΔT (°C)/Δt (min); T2, T3, T4 are the temperatures (°C) of the cold base (below freezing point); T1 (°C) is the hot base temperature (above freezing point); V is the velocity (mm/s) at which the sample moves from T1 to T4; G1, G2, G3 are the temperature gradients (°C/mm).

## Data Availability

Not applicable.

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
