# Peer review of "Cryopreservation by Directional Freezing and Vitrification Focusing on Large Tissues and Organs"

_cells, 2022, doi:10.3390/cells11071072_

Round 1
Reviewer 1 Report
The revised paper is now very clear, understandable, I have only one hint for correction:
- page 4: the equation for the probability of fracture and for the probability of vitrification should be very similarly presented, and as follows: Pfrac and Pvitr
Author Response
Thank you very much, I did changed as suggested
Reviewer 2 Report
This manuscript covers a number of aspects in tissue and organ cryopreservation. This information can help scientist to understand the evolution and mechanism of both conventional freezing and vitrification.Author Response
Thank you for your comment
Reviewer 3 Report
The author submitted a review paper on the cryopreservation of whole-organ and tissue. It touches on the important aspect of cryobiology. This is my second time reviewing this paper. The quality and readability of this paper are very much improved. The discussion on the cryobiology principles of large tissues and organs cryopreservation is insightful and thorough. Before final publication, some slight revisions are needed.
I understand the control of ice growth patterns is difficult especially in samples with large volumes, same as the control of latent heat release. I guess this is the major advantage of directional freezing? But, the explanations about how directional freezing working is still very hard to follow, my questions are basically focused on the interpretation of figure 1:
- The drawing of figure 1 is still relatively simple, the author may carefully design and redraw that figure with higher quality.
- Why the ice growth strategy is different for the freezing of cells and organs?
- How directional freezing inhibits the initial sample supercooling?
Besides, I enjoy the author’s way of using mathematics to study biological issues.
Author Response
I would like to thank the reviewer for the second time..
- The drawing of figure 1 is still relatively simple, the author may carefully design and redraw that figure with higher quality. I tried to draw a figure in a way that the differences between organ freezing and cells freezing will be clear, I dont think that higher quality will explain better these differences.
- Why the ice growth strategy is different for the freezing of cells and organs?- In general you right, the strategy should not be different, but because homogeneus cooling rate in large volume is more difficult to achieve, we should reduce velocity (V) of the tube movement to slower rate than the rate of heat transfer from the center of the organ to the conductive block which surround it .
- How directional freezing inhibits the initial sample supercooling? when we start the freezing process, the tip of the tube is touching the cold block (T2) and ice crystals are formed, following the movement of the tube, the ice crystals grow and follow the freezing point of the solution (continuous seeding) which inhibit the possible supercooling of the sample.
Reviewer 4 Report
The review summarizes the approaches for whole organ cryopreservation through using two ways: directional freezing and vitrification.
There are some comments that should be carefully revised.
1- Please add a subheading for "limitations and challenges".
2- Author contribution and conflict of interest should be separately described.
3- You focused on oocytes and embryos, while they are not whole organs. Please include this in the title and in the abstract.
4- The earlier works of G. Vajta regarding oocyte/embryo vitrification are not cited.
5- How about ovarian tissue vitrification?
6- Clinical trials (https://www.clinicaltrials.gov/) for these approaches should be also included in the review.
Author Response
1- Please add a subheading for "limitations and challenges". I added the subheading
2- Author contribution and conflict of interest should be separately described. I separate between the two
3- You focused on oocytes and embryos, while they are not whole organs. Please include this in the title and in the abstract. I changed the title.
4- The earlier works of G. Vajta regarding oocyte/embryo vitrification are not cited. In a separate review on oocytes and embryos vitrification (37), I cited the work of Vajta several times.
5- How about ovarian tissue vitrification? I currently work on a new method for ovarian tissue vitrification which will be publish soon.
6- Clinical trials (https://www.clinicaltrials.gov/) for these approaches should be also included in the review. I dont see how it can be incorporate in this review, but thank you for showing me this excellent website.
Round 2
Reviewer 4 Report
The manuscript has been corrected.
This manuscript is a resubmission of an earlier submission. The following is a list of the peer review reports and author responses from that submission.
Round 1
Reviewer 1 Report
In the revised manuscript, I found corrections done to my questions and hints.
However, Authors changed the character of the paper, and turned it to more experimental than the review paper. To make the situation clear, I would advise to:
- remove the theoretical presentation of principles of freezing and vitrification,
- leave the experimental part with the appropriate discussion and conclusions,
- finally - change the title ie leave the second part of the previous version,
- prepare a new review paper on theoretical aspects of cryopreservation of the whole organ/tissues.